# Look where you look! Saliency-guided Q-networks for generalization in visual Reinforcement Learning

**David Bertoin**†
IRT Saint-Exupéry
ISAE-SUPAERO
IMT, INSA Toulouse
ANITI
Toulouse, France
`david.bertoin@irt-saintexupery.com`

**Adil Zouitine**†
IRT Saint-Exupéry
ISAE-SUPAERO
Toulouse, France
`adil.zouitine@irt-saintexupery.com`

**Mehdi Zouitine**†
IRT Saint-Exupéry
IMT, Université Paul Sabatier
Toulouse, France
`mehdi.zouitine@irt-saintexupery.com`

**Emmanuel Rachelson**†
ISAE-SUPAERO
Université de Toulouse
ANITI
Toulouse, France
`emmanuel.rachelson@isae-supaero.fr`

## Abstract

Deep reinforcement learning policies, despite their outstanding efficiency in simulated visual control tasks, have shown disappointing ability to generalize across disturbances in the input training images. Changes in image statistics or distracting background elements are pitfalls that prevent generalization and real-world applicability of such control policies. We elaborate on the intuition that a good visual policy should be able to identify which pixels are important for its decision, and preserve this identification of important sources of information across images. This implies that training of a policy with small generalization gap should focus on such important pixels and ignore the others. This leads to the introduction of saliency-guided Q-networks (SGQN), a generic method for visual reinforcement learning, that is compatible with any value function learning method. SGQN vastly improves the generalization capability of Soft Actor-Critic agents and outperforms existing state-of-the-art methods on the Deepmind Control Generalization benchmark, setting a new reference in terms of training efficiency, generalization gap, and policy interpretability.

## 1 Introduction

Deploying reinforcement learning (RL) [Sutton and Barto, 2018] algorithms in real-life situations requires overcoming a number of still open challenges. Among these is the ability for the trained control policies to focus their attention on causal state features and ignore confounding factors [Machado et al., 2018, Henderson et al., 2018]. In visual RL tasks, this implies for instance being able to ignore the background and other distracting factors, even when they might be somehow correlated with progress within the task at hand. Despite a very active trend of research on the topic of closing the generalization gap for RL agents [Cobbe et al., 2019, 2020, Song et al., 2019, Hansen et al., 2021, Hansen and Wang, 2021], current algorithms are still rather brittle when it comes to filtering out such distracting factors, which hinders their applicability to real-life scenarios.

---

†These authors contributed equally to this work

36th Conference on Neural Information Processing Systems (NeurIPS 2022).

In the present work, we propose a novel method which encourages the agent to identify efficiently crucial input pixels, and strengthen the policy's dependency on those pixels. In plain words, we encourage the agent to pay attention to, and be self-aware of where it looks in input images, in order to make its decision policy more focused on important areas and less sensitive to ambiguous or distracting pixels. This intention is expressed within the generic method of saliency-guided Q-networks (SGQN), which can be applied on any approximate value iteration based, deep RL algorithm. SGQN relies on two core mechanisms. First, it regularizes the value function learning process with a consistency term that encourages the value function to depend in priority on pixels that are identified as decisive. The second mechanism pushes the agent to be self-aware of which pixels are responsible for making decisions, and encode this information within the extracted features. This second mechanism translates to a self-supervised learning objective, where the agent trains to predict its own Q-value's saliency maps. In turn, this improves the regularization of the value function learning phase, which provide better labels for the self-supervised learning phase, overall resulting in a virtuous improvement circle.

SGQN is a simple, generic method, that permits many variants in the way the two core mechanisms are implemented. In the present paper, we demonstrate that applying SGQN to soft actor-critic agents [Haarnoja et al., 2018] dramatically enhances their quality on the DMControl generalization benchmark [Hansen and Wang, 2021], a standard evaluation benchmark for generalization in continuous actions RL. SGQN already improves the training efficiency of such agents in domains without distractions. But most importantly, it sets a new state-of-the-art in terms of generalization performance, in particular in especially difficult benchmarks where previous methods suffered from confounding factors. As a side benefit, it also provides explanations of its own decisions at run time, under the form of interpretable attribution maps, with no overhead cost and no need to compute ad hoc saliency maps, which is another desirable property in the pursuit of deployable RL.

Section 2 of this paper introduces the necessary background and state-of-the-art in closing the generalization gap in RL, as well as attribution methods, leading to the key intuitions underpinning SGQN. Section 3 introduces the method itself and implements it within soft actor-critic agents. Section 4 evaluates SGQN's training efficiency, generalization capabilities, and policy interpretability. It also discusses the different design choices made along the way and some foreseeable limitations. Section 5 summarizes and concludes this paper.

## 2   Background and related work

**Reinforcement learning (RL).** RL [Sutton and Barto, 2018] considers the problem of learning a decision making policy for an agent interacting over multiple time steps with a dynamic environment. At each time step, the agent and environment are described through a state $s \in \mathcal{S}$, and an action $a \in \mathcal{A}$ is performed; then the system transitions to a new state $s'$ according to probability $\mathcal{P}(s'|s,a)$, while receiving reward $\mathcal{R}(s,a)$. The tuple $M = (\mathcal{S}, \mathcal{A}, \mathcal{P}, \mathcal{R})$ forms a Markov Decision Process (MDP) [Puterman, 2014], which is often complemented with the knowledge of an initial state distribution $p_0(s)$. A decision making policy parameterized by $\theta$ is a function $\pi_\theta(a|s)$ mapping states to distributions over actions. Training a reinforcement learning agent consists in finding the policy that maximizes the discounted expected return $J(\pi_\theta) = \mathbb{E}[\sum_{t=0}^{\infty} \gamma^t \mathcal{R}(s_t, a_t)]$.

**Poor generalization in RL.** Despite the recent progress of (deep) RL algorithms in solving complex tasks, a number of studies have pointed out their poor generalization capabilities. Using a grid-world maze environment, Zhang et al. [2018c] demonstrate the ability of deep RL agents to memorize a non-trivial number of training levels with completely random rewards. Using attribution methods, Song et al. [2019] highlight what they define as *observational overfitting* i.e., the propensity of RL agents to base their decision on background uninformative elements observed during training, instead of the semantic pieces of information one could intuitively expect such as object positions or relations. Zhang et al. [2018a] measure the generalization error in continuous control environments by training and testing agents on different sets of seeds. Zhao et al. [2019] define generalization in RL as robustness to a distribution of environments, and samples environments from this distribution to learn a robust policy. Overall, these works illustrate the lack of generalization abilities of vanilla deep RL algorithms, either to states that were not encountered during training, or to variations in the transition dynamics. In the present work, we aim to shape the policy learning process, so that it relies on meaningful features that permit such generalization and robustness.

**Evaluating generalization in RL.** Under the impetus of these works and the need for benchmarks with separate training and testing environments [Whiteson et al., 2011, Machado et al., 2018, Henderson et al., 2018], original benchmarks for evaluating the generalization capacities of RL agents have been designed. Packer et al. [2018] propose a modified version from control problems in OpenAI Gym [Brockman et al., 2016] and Roboschool [Schulman et al., 2017] that lets the user change the system dynamics. Machado et al. [2018] propose a modified version of the ALE environments [Bellemare et al., 2013] allowing one to change modes and difficulties. Without modifying the underlying transition model, Zhang et al. [2018b], Grigsby and Qi [2020], Stone et al. [2021] add distracting elements (e.g., addition of real images or videos in the background, change of colors) to the ALE environments and the Deepmind control suite. Even if the modifications to the original environments do not alter the semantic information, they already appear to be challenging for agents prone to observational overfitting. Cobbe et al. [2019, 2020], Juliani et al. [2019] use procedural content generation to design highly randomized sets of environments with different level layouts, game assets, and objects locations, letting the user study robustness to several independent variation factors. One may note that the diversification of learning environments is in itself a first practical way to induce generalization [Tobin et al., 2017, Cobbe et al., 2019, 2020] and also permits curriculum-based learning [Jiang et al., 2021]. Nevertheless, when the diversity of training scenarios is lacking, three sets of methods are generally employed, as detailed below.

**Regularization.** Farebrother et al. [2018], Cobbe et al. [2019, 2020] demonstrated the beneficial effects of popular regularization methods from the supervised learning literature. Igl et al. [2019] mitigate the adverse effect that classical regularization may have on the gradient quality with selective noise injection and combine it with an information bottleneck regularization. Inspired by mixup [Zhang et al., 2018d], Wang et al. [2020] use mixtures of observations to stimulate linearity in the policy's outputs in-between states.

**Data augmentation.** Laskin et al. [2020a] evidence the benefits of training RL agents with augmented data (RAD). Yarats et al. [2020] average both the value function and its target over multiple image transformations (DrQ). Hansen et al. [2021] only apply data augmentation in $Q$-value estimation without augmenting $Q$-targets used for boostrapping. Raileanu et al. [2021] combine the previous method with UCB [Auer, 2002] to pick the most promising augmentation, and apply it to PPO [Schulman et al., 2017]. Yuan et al. [2022] propose a task-aware Lipschitz data augmentation method (TLDA) to augment task irrelevant pixels. Fan et al. [2021] use weak data augmentation to train an expert without hindering its performance and distill its policy to a student trained with substantial data augmentation. Besides augmentations of raw inputs, other augmentations operate directly within the agents' network. Lee et al. [2020] introduce a random convolutions layer at the earliest level of the agent's network to modify the *texture* of the visual observations. Zhou et al. [2020] adapt mixup [Zhang et al., 2018d] with style statistics encoded in early instance normalization layers to increase data diversity. Bertoin and Rachelson [2022] apply channel-consistent local permutations of the feature map to induce robustness to spatial spurious correlations. Finally, data augmentation can also be used in an auxiliary loss to promote invariance to distributional shift in representations. Hansen and Wang [2021] propose a soft-data augmentation method (SODA) by adding an auxiliary self-supervised learning phase to SAC [Haarnoja et al., 2018], similar to BYOL [Grill et al., 2020].

**Representation learning.** Higgins et al. [2017b] demonstrate zero-shot adaptation to unseen configurations in testing environments, using a $\beta$-VAE [Higgins et al., 2017a] to learn disentangled representations. Fan and Li [2021] jointly maximize the mutual information between sequences of observations to remove the task-irrelevant information. Fu et al. [2021] learn a disentangled world model that separates reward-correlated features from background. Wang et al. [2021] extract, using visual attention, the observation foreground to provide background invariant inputs to the policy learner. Raileanu and Fergus [2021] separate the actor from the critic in the agent's network architecture and add an adversarial auxiliary objective on the actor's representations to remove the information needed to estimate the value function that is not irrelevant to a general policy. Zhang et al. [2020] train an encoder to project states so that their distances match with the bisimulation distances in state space. Other recent works use behavioral similarities combined with contrastive learning [Agarwal et al., 2020] or clustering [Mazoure et al., 2022] to map behaviorally similar observations to similar representations.

**Attributing decisions to inputs.** Although not directly aiming at generalization, a related topic is that of *attribution*, where one wishes to identify which parts of an input are responsible for major changes in the output of a function. Intrinsically, computing attributions boils down to computing (some trans-

formation of) the gradient of the function's output with respect to the input's components. Computational graphs of differentiable functions, such as neural networks, are particularly suited to computing attributions by using the back propagation algorithm [Simonyan et al., 2014, Springenberg et al., 2015, Smilkov et al., 2017, Selvaraju et al., 2017, Chattopadhay et al., 2018]. When these methods are applied to images, one obtains a map which is known as a *saliency map* or *attribution map*. Such attribution maps indicate which input pixels are determining for a policy's output (or the Q-value of action $a$) and thus permit interpretation of the function itself, rather than its point-wise decision alone. Mousavi et al. [2016], Greydanus et al. [2018], Atrey et al. [2020] have used saliency maps to analyze and explain the behavior of RL agents. Rosynski et al. [2020] indicates in particular that guided backpropagation [Springenberg et al., 2015] provides good visualizations of RL policies across a span of environments. Most existing works, however, only exploit saliency maps as tools for interpretation. Closely related to our contribution is that of Ismail et al. [2021], who incorporate attributions into their training process in supervised learning. Their procedure iteratively uses a binary mask computed from attributions to remove features with small and potentially noisy gradients while maximizing the similarity of model outputs for both masked and unmasked inputs. This saliency guided regularization improves the quality of gradient-based saliency explanations without interferring with training stability.

**This contribution.** The rationale of the method we introduce in the next section is to encourage the agent to generalize to new states, based on which pixels are identified as important decision factors. For this purpose, we perform pixel-level masking on the input image, depending on the computed attribution, and regularize the value function learning process with the difference in Q-values. This way, we encourage the value function to focus specifically on the pixels with high attribution. Leveraging data augmentation, self-supervised learning methods have demonstrated their ability to induce features that are insensitive to lighting, background, and high-frequency noise. The method we propose does not directly use data augmentation in the value function or policy update phase. Instead, it is introduced during an auxiliary phase where the augmented state's encoding is used to predict the attribution mask of the original state. This way, the encoder is encouraged to preserve information that is useful for predicting which pixels were important in the agent's decision-making. A parallel with SODA can thus be made by considering that the projector used in the BYOL objective is here replaced by a surrogate of the derivative of the value function, allowing to refine the quality of the projection and to learn which pixels and visual features are consistently important across states to predict the Q-values. The value function regularization and the self-supervised learning objective are mutually beneficial: the former outputs sharper saliency maps from the value function, that serve as better labels for the auxiliary self-supervised learning task, which in turn induces better features and better attributions. In short, we encourage the agent to pay attention to where it is looking, with the intention that this triggers more efficient learning and more interpretable output.

## 3 Saliency-guided Q-networks

We propose a generic saliency-guided Q-networks (SGQN) method, for visual deep reinforcement learning. In a nutshell, SGQN considers the application of the binarized attribution map as a mask over the input state and regularizes the value function learning objective with a consistency term between the Q-values of the masked and the original state images. It also defines an auxiliary self-supervised learning task that aims to match a prediction of the attribution map on an augmented image, with the attribution map of the original image. Such an auxiliary task orients the gradient descent towards features that are shared across states, as illustrated by the work of Hansen and Wang [2021]. SGQN can be combined with any value function learning objective, any attribution map computation technique, any image augmentation method for self-supervised learning, and is suited for both discrete and continuous actions. We first present SGQN as a generic enhancement of approximate value iteration methods. Then we derive a specific version built on SAC [Haarnoja et al., 2018] and on the guided backpropagation algorithm [Springenberg et al., 2015].

A vast number of deep RL algorithms belong to the family of approximate value iteration methods. Such methods build a sequence of $(Q_n)_{n\in\mathbb{N}}$ and $(\pi_n)_{n\in\mathbb{N}}$ functions that aim to asymptotically tend to $Q^*$ and $\pi^*$. $Q_{n+1}$ is defined as a minimizer of $L_Q = \|Q - T^{\pi_n}Q_n\|$, where $T^{\pi_n}$ is the Bellman evaluation operator with respect to $\pi_n$. Then $\pi_{n+1}$ is defined by applying a *greediness* operator $\mathcal{G}$ to $Q_{n+1}$ and the process is iterated. Geist et al. [2019] showed how one could introduce regularization within the expression of $L_Q$, yielding the class of regularized MDPs. The classical DQN algorithm [Mnih et al., 2015] approximates the solution to $L_Q$ by taking a number of gradient steps with respect

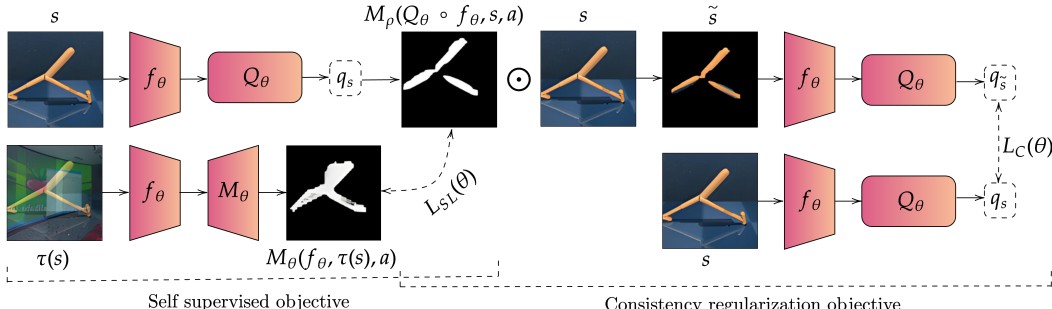

Figure 1: SGQN losses. The $L_{SL}$ self-supervised loss trains $f_\theta$ so that $M_\theta(f_\theta(\tau(s)), a)$ predicts $M_\rho(Q_\theta \circ f_\theta, s, a)$. In turn, the $L_C$ consistency loss pushes $Q_\theta \circ f_\theta$ to only depend on salient pixels.

to a target network $Q_n$ and uses an $\arg\max$ greediness operator. When actions are continuous, actor-critic methods introduce a surrogate model of the $Q_{n+1}$-greedy policy, under the form of an actor network $\pi_{n+1} = \mathcal{G}(Q_{n+1})$ obtained by gradient ascent. In what follows, we denote by $L_Q(\theta)$ the loss minimized by a generic learning procedure for $Q_\theta$, based on the Bellman operator, independently of whether it is regularized, uses double critics, etc. Similarly, we note $L_\pi(\theta)$ the loss minimized to yield a greedy policy $\pi_\theta$, when applicable.

We denote by $M(Q, s, a)$ an attribution map for $Q(s, a)$, in the space of images $\mathcal{S}$. Vanilla grad [Simonyan et al., 2014] for instance will compute such a map under the form $M(Q, s, a) = \partial Q(s, a)/\partial s$, while guided backpropagation [Springenberg et al., 2015] will mask out negative gradients, yielding a different attribution map. We note $M_\rho(Q, s, a)$ the binarized value attribution map where $M_\rho(Q, s, a)_j = 1$ if attribution pixel $M(Q, s, a)_j$ belongs to the $\rho$-quantile of highest values for $M(Q, s, a)$, and 0 otherwise.

The proposed method is built on a classical $Q$-network architecture. The value function is divided into 2 parts: an encoder $f_\theta : \mathcal{S} \to \mathcal{Z}$ and a $Q$-function $Q_\theta : \mathcal{Z} \times \mathcal{A} \to \mathbb{R}$ built on top of this encoder. We add a decoder function $M_\theta$ after the features encoder $f_\theta$, such that $M_\theta(f_\theta(s), a)$ aims to predict the attribution map of $Q_\theta(f_\theta(s), a)$. Many algorithms require defining double critics [Fujimoto et al., 2018] or target networks $f_\psi$ and $Q_\psi$ which are often updated with an exponential moving average of $\theta$ [Polyak and Juditsky, 1992]. We omit them here for clarity, although their introduction in SGQN is straightforward. When needed, a policy head $\pi_\theta : \mathcal{Z} \to \mathcal{A}$ is built on top of the encoder $f_\theta$ to define the actor network. The backbone architecture and training process are summarized in Figure 1. The SGQN training procedure involves two additional objectives: a consistency objective responsible for regularizing the critic update and an auxiliary supervised learning objective.

**The consistency regularization objective** (Figure 1 right) $L_C(\theta) = \mathbb{E}_{s,a}[[Q_\theta(f_\theta(s), a) - Q_\theta(s \odot M_\rho(Q_\theta \circ f_\theta, s, a), a)]^2]$ (where $\odot$ denotes the Hadamard product), is added to the classical critic loss $L_Q(\theta)$ during the critic update phase. This loss function encourages the Q-network $Q_\theta \circ f_\theta$ to make its decision based in priority on the salient pixels in $M(Q, s, a)$, hence promoting consistency between the masked and original images. The new critic objective function is thus defined as $L_Q(\theta) + \lambda L_C(\theta)$.

**The self-supervised learning phase** (Figure 1 left) updates the parameters of $f_\theta$ so that given a generic image augmentation function $\tau$, $(f_\theta(\tau(s)), a)$ contains enough information to accurately reconstruct the attribution mask $M_\rho(Q_\theta, s, a)$. This defines a self-supervised learning objective function $L_{SL}(\theta) = \mathbb{E}_{s,a}[BCE(M_\theta(f_\theta(\tau(s)), a), M_\rho(Q_\theta, s, a)]$, where $BCE$ is the binary cross entropy loss, which could be replaced by any other measure of discrepancy between attribution maps.

The interplay between these two phases acts as a virtuous circle. The consistency regularization loss, similar to that of Ismail et al. [2021], pushes the network to focus its decision on a selected set of pixels (hence relying on the assumption that initial saliency maps are reasonably good). This enhances the contrast between pixels in the gradient image, and thus yields sharp saliency maps, even before binarization. These maps serve as a target labels during the self-supervised learning phase; since they are less noisy than without the consistency loss, they provide a stronger incentive to encode the information of which pixels are important, within $f_\theta$. In turn, as exemplified by Hansen and Wang [2021] and Grill et al. [2020], the features obtained by the self-supervised learning procedure

tend to be insensitive to background, noise, or exogenous conditions, and provide features that are shared across observations. These features benefit from the better labels (less irrelevant pixels in the attribution map). Finally, this helps provide good pixel attributions that will be used in the minimization of the consistency loss during the critic phase. Appendix G proposes an extended discussion on this virtuous circle.

Note that the thresholding operation performed to obtain $M_\rho$ is not strictly necessary, either in the consistency loss or in the self-supervised learning one. Instead of a hard thresholding, one could turn to a normalization of the attribution map, such as a softmax for instance. Such a soft-attribution image remains fully compatible with SGQN. The choice to keep the thresholded $\rho$-quantile mask is motivated by the arguments of Ismail et al. [2021] who extensively study such variations and conclude to the benefits of this binarized mask. One could also remark that SGQN does not require to use the target network during the self-supervised learning phase, which contrasts with the choices of SODA or BYOL and makes the method somewhat more versatile.

Algorithm 1 presents the pseudo-code of combining SGQN with SAC, yielding an SG-SAC algorithm. Note that, for the sake of simplicity, we write $\theta$ for the full set of network parameters, which are thus shared by the encoder, the Q-value head, the policy head, and the attribution reconstruction head.

---

**Algorithm 1:** Saliency-guided SAC (changes to SAC in blue)

**Parameters:** frequency of auxiliary updates $N_{SL}$, attribution quantile value $\rho$, learning rate $\alpha$, data augmentation function $\tau$.

**for** *each interaction time step* **do**

$\quad a, s' \sim \pi_\theta(\cdot \mid f_\theta(s)), \mathcal{P}(\cdot \mid s, a)$                 `// Sample a transition`

$\quad \mathcal{B} \leftarrow \mathcal{B} \cup \{(s, a, \mathcal{R}(s, a), s')\}$          `// Add transition to replay buffer`

$\quad \{s_i, a_i, r(s_i, a_i), s'_i\}_{i \in [1, N]} \sim \mathcal{B}$     `// Sample a mini-batch of transitions`

$\quad \theta \leftarrow \theta - \alpha \nabla_\theta L_Q(\theta) + \lambda L_C(\theta)$               `// Critic update phase`

$\quad \theta \leftarrow \theta - \alpha \nabla_\theta L_\pi(\theta)$                      `// Actor update`

$\quad$ Every $N_{SL}$ steps: $\theta \leftarrow \theta - \alpha \nabla_\theta L_{SL}(\theta)$       `// Self-supervised learning`

Note: $L_Q$ and $L_\pi$ are as defined by Haarnoja et al. [2018], temperature update, double critics and target network updates are omitted here for clarity.

---

## 4   Experimental results and discussion

This section evaluates SGQN's training efficiency, generalization capabilities, and policy interpretability. It also discusses the different design choices made along the way. We compare our approach with current state-of-the-art methods for generalization in continuous actions RL (RAD [Laskin et al., 2020b], DrQ [Yarats et al., 2020], SODA [Hansen and Wang, 2021], SVEA [Hansen et al., 2021]) on five environments from the DMControl Generalization Benchmark (DMControl-GB) [Hansen and Wang, 2021]. The DMControl-GB presents a variety of vision-based continuous control tasks based on the Deepmind control suite. Agents are trained in a fixed background environment and evaluated under two challenging distribution shifts, consisting in replacing the training background with natural videos. Figure 2 illustrates the effects of both domain shifts. All the compared methods herein are variants of SAC, for which we use the same architecture for all agents. These methods all use data augmentation in one of their stages. Following the experimental protocol of the competing approaches, we used a random overlay augmentation [Hansen and Wang, 2021] (consisting in blending together original observations with random images from the Places365 dataset [Zhou et al., 2017]) for all the methods except for RAD and DrQ, for which we used random crops and random shifts respectively, as it is reported as producing the best results [Laskin et al., 2020b, Yarats et al., 2020]. We trained all agents for $500\,000$ steps using the vanilla training environment with no visual variation. Appendix A, B, and E discuss all the hyperparameters, network architectures, and implementation choices used for this benchmark. Appendix C includes extra experimental results on DMControl-GB and Appendix D provides additional results on a vision-based robotic environment.

**SGQN improves value iteration in the training domain.** We first compare the performance in the training domain, with no visual distractions, of SAC and SGQN, on five environments from the DMControl-GB (Figure 3). The SGQN agents outperform, by a considerable margin, the SAC agents both in terms of asymptotic performance and sample efficiency on 4 out of the 5 environments. In

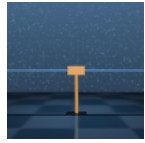 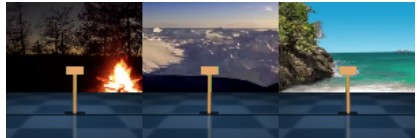 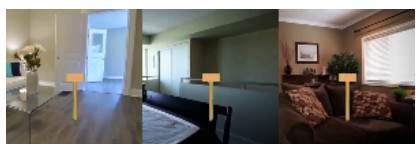

|              (a) Training              |      (b) *Video easy* distribution shift      |      (c) *Video hard* distribution shift      |

Figure 2: Examples of training and testing environments

| Benchmark | Environment | SAC | DrQ | RAD | SODA | SVEA | SGQN | Δ |
|---|---|---|---|---|---|---|---|---|
| Easy | Walker walk | 245 ± 165 | 747 ± 21 | 608 ± 92 | 771 ± 66 | 828 ± 66 | **910 ± 24** | **+82(10%)** |
| | Walker stand | 389 ± 131 | 926 ± 30 | 879 ± 64 | 965 ± 7 | **966 ± 5** | 955 ± 9 | −11(1%) |
| | Ball in cup | 192 ± 157 | 380 ± 188 | 363 ± 158 | 939 ± 10 | 908 ± 55 | **950 ± 24** | **+11(1%)** |
| | Cartpole | 398 ± 60 | 459 ± 81 | 473 ± 54 | 742 ± 73 | 753 ± 45 | **761 ± 28** | **+8(1%)** |
| | Finger spin | 206 ± 169 | 599 ± 62 | 516 ± 113 | 783 ± 51 | 723 ± 98 | **956 ± 26** | **+173(22%)** |
| | **Average** | 286 | 622 | 568 | 836 | 836 | **906** | **+70(8%)** |
| Hard | Walker walk | 122 ± 47 | 121 ± 52 | 80 ± 10 | 312 ± 32 | 385 ± 63 | **739 ± 21** | **+354(92%)** |
| | Walker stand | 231 ± 57 | 252 ± 57 | 229 ± 45 | 736 ± 132 | 747 ± 43 | **851 ± 24** | **+104(14%)** |
| | Ball in cup | 101 ± 37 | 100 ± 40 | 98 ± 40 | 381 ± 160 | 498 ± 174 | **782 ± 57** | **+284(57%)** |
| | Cartpole | 158 ± 17 | 136 ± 29 | 152 ± 29 | 403 ± 17 | 401 ± 38 | **544 ± 43** | **+141(35%)** |
| | Finger spin | 13 ± 10 | 38 ± 13 | 39 ± 20 | 309 ± 49 | 307 ± 24 | **822 ± 24** | **+513(166%)** |
| | **Average** | 125 | 129 | 119 | 430 | 468 | **748** | **+280(60%)** |

Table 1: Performance on *video easy* and *video hard* testing levels. Δ = difference with second best.

addition to obtaining better results, the variance of the agents trained with SGQN is also significantly lower than that of the agents trained with SAC, demonstrating that the enhancements employed in SGQN have a beneficial effect on the stability of the training, regardless of the ability to generalize across domains.

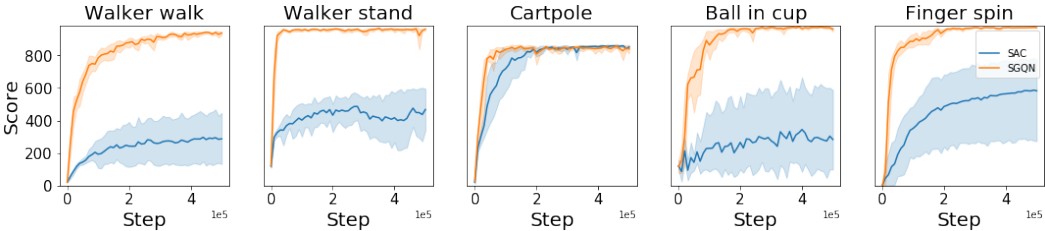

Figure 3: Comparison of SAC and SGQN training learning curves

**SGQN improves generalization.** We assess the zero-shot generalization ability of SGQN on the *video easy* and *video hard* benchmarks from the DMControl-GB. The easy version only replaces the background of the training image with a distracting image, while the hard version also replaces the ground and the shadows (Figure 2). We report the average sum of rewards after $500\,000$ training steps for the *video easy* benchmark in the top part of Table 1. Agents trained with SGQN outperform agents trained with other state-of-the-art methods on all tasks but one (where it is on par with other agents), thus demonstrating the generalization capabilities induced by the method. By removing the ground and shadows, the *video hard* benchmark (bottom part of Table 1) causes a larger, more confusing, and more challenging distributional shift. All the competitors of SGQN experience a radical decrease in their generalization performance. SGQN is significantly less impacted and outperforms all its competitors with an average margin over the second-best of 60% on all environments, and a gain range of 14 to 166% across environments. Figure 4 reports the evolution of each agent's score on the *video hard* environments, along training. SODA and SVEA's scores drop drastically when the ground and shadows are removed. SGQN is less sensitive to this change, notably through the consistency loss, which encourages agents to make decisions based on the subsets of pixels they deem most interesting. Overall, the interplay of the two phases of SGQN seems to be key to a major leap forward in terms of generalization gap in difficult environments, setting a new reference state-of-the-art.

**SGQN yields sharp saliency maps.** We use guided backpropagation to visually compare the SGQN agents' ability to discriminate the essential information with that of other agents. Figure 5 shows an example of the binarized attribution maps for all agents in a *video hard* state. While the other agents

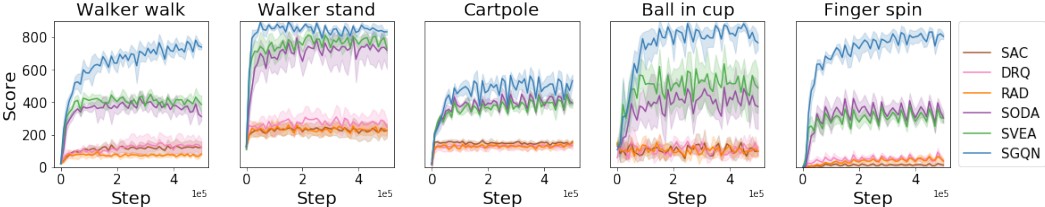

Figure 4: Performance on *video hard* testing levels.

seem to be disturbed by background elements and retain some dependency on background pixels in their decision, the attributions of the agent trained with SGQN are precisely located on the important information, hence suggesting better generalization potential.

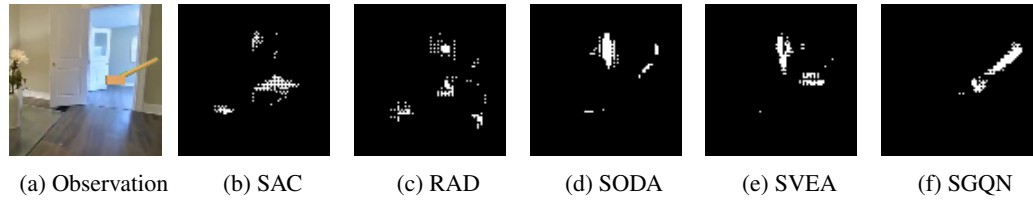

(a) Observation    (b) SAC    (c) RAD    (d) SODA    (e) SVEA    (f) SGQN

Figure 5: Example of attributions in *video hard* levels

**SGQN is interpretable by design.** In all our experiments, we trained the SGQN agents using guided backpropagation. We emphasize that any other attribution method could be used instead. Some of these methods are expensive and require several forward (or backward) passes within the network (e.g., RISE [Petsiuk et al., 2018], or the work of Fel et al. [2021]) to yield attribution maps which explain the agent's decision. In contrast, SGQN's auxiliary phase trains a predictor to estimate the most important pixels according to the chosen attribution method. Therefore, this predictor is a surrogate of the attribution method used during training. It allows identifying the essential image features that condition the agent's decision in the same forward pass as the prediction of the action itself, without incurring the cost of one or several costly additional backward or forward passes. Figure 6 illustrates the proximity between the actual saliency map and the attribution surrogate model $M_\theta$. This makes SGQN both self-aware (of its own saliency maps) and intrinsically interpretable from a human perspective, with no computational overhead at evaluation time.

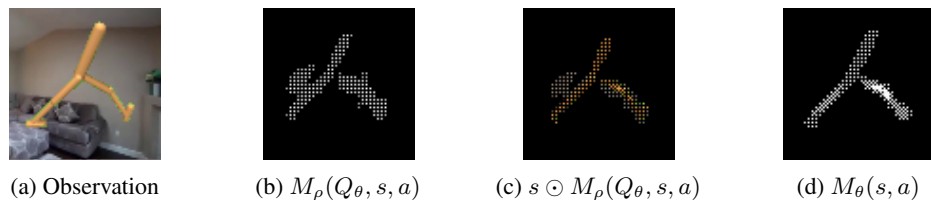

(a) Observation    (b) $M_\rho(Q_\theta, s, a)$    (c) $s \odot M_\rho(Q_\theta, s, a)$    (d) $M_\theta(s, a)$

Figure 6: Comparison between the true saliency map (b), the masked image (c) used in the consistency regularization term, and the estimated saliency map (d) in a *video hard* level.

**Ablation study.** SGQN relies on two enhancements of vanilla approximate value iteration: the auxiliary self-supervised learning phase and the consistency regularization term in the critic's loss. We perform an ablation study to assess their individual contribution and Table 2 reports the average sum of rewards in the training domain and for zero-shot generalization in all environments (Appendix F takes a different perspective and compares SGQN with the combination of SVEA and SODA). Individually, each of these features greatly improves both training and zero-shot generalization performance on all environments. The auxiliary self-supervised learning phase provides the most significant performance gains over vanilla SAC. The average training performance on all environments of agents trained with this auxiliary objective is more than 73% higher than that of agents trained with vanilla SAC. The same applies to the performance in zero-shot generalization, which increases by more than 146% on *video easy* and by more than 198% on *video hard* environments. One can note

| Environment | benchmark | SAC | SAC+Consistency | SAC+Self learning | SGQN |
|---|---|---|---|---|---|
| Walker walk | train | $287 \pm 165$ | $449 \pm 100 \, (\mathbf{+56}\%)$ | $934 \pm 28 \, (\mathbf{+225}\%)$ | $937 \pm 12 \, (\mathbf{+226}\%)$ |
| | easy | $245 \pm 165$ | $423 \pm 96 \, (\mathbf{+73}\%)$ | $844 \pm 53 \, (\mathbf{+244}\%)$ | $910 \pm 24 \, (\mathbf{+271}\%)$ |
| | hard | $122 \pm 47$ | $344 \pm 87 \, (\mathbf{+182}\%)$ | $226 \pm 48 \, (\mathbf{+85}\%)$ | $739 \pm 21 \, (\mathbf{+505}\%)$ |
| Walker stand | train | $467 \pm 162$ | $857 \pm 120 \, (\mathbf{+84}\%)$ | $957 \pm 11 \, (\mathbf{+105}\%)$ | $960 \pm 9 \, (\mathbf{+106}\%)$ |
| | easy | $389 \pm 131$ | $846 \pm 107 \, (\mathbf{+117}\%)$ | $944 \pm 14 \, (\mathbf{+143}\%)$ | $955 \pm 9 \, (\mathbf{+145}\%)$ |
| | hard | $231 \pm 57$ | $696 \pm 150 \, (\mathbf{+201}\%)$ | $769 \pm 32 \, (\mathbf{+233}\%)$ | $851 \pm 24 \, (\mathbf{+268}\%)$ |
| Ball in cup | train | $284 \pm 329$ | $755 \pm 261 \, (\mathbf{-8}\%)$ | $967 \pm 1 \, (\mathbf{+240}\%)$ | $971 \pm 7 \, (\mathbf{+242}\%)$ |
| | easy | $192 \pm 157$ | $440 \pm 214 \, (\mathbf{+129}\%)$ | $705 \pm 43 \, (\mathbf{+267}\%)$ | $950 \pm 24 \, (\mathbf{+399}\%)$ |
| | hard | $101 \pm 37$ | $190 \pm 63 \, (\mathbf{+88}\%)$ | $203 \pm 122 \, (\mathbf{+100}\%)$ | $782 \pm 57 \, (\mathbf{+670}\%)$ |
| Cartpole | train | $850 \pm 29$ | $863 \pm 9 \, (\mathbf{+2}\%)$ | $857 \pm 20 \, (\mathbf{+1}\%)$ | $839 \pm 37 \, (\mathbf{-1}\%)$ |
| | easy | $398 \pm 60$ | $663 \pm 96 \, (\mathbf{+67}\%)$ | $647 \pm 42 \, (\mathbf{+63}\%)$ | $761 \pm 28 \, (\mathbf{+91}\%)$ |
| | hard | $158 \pm 17$ | $228 \pm 62 \, (\mathbf{+44}\%)$ | $294 \pm 40 \, (\mathbf{+86}\%)$ | $544 \pm 43 \, (\mathbf{+244}\%)$ |
| Finger spin | train | $829 \pm 21$ | $985 \pm 1 \, (\mathbf{+19}\%)$ | $985 \pm 1 \, (\mathbf{+19}\%)$ | $985 \pm 1 \, (\mathbf{+19}\%)$ |
| | easy | $382 \pm 40$ | $865 \pm 65 \, (\mathbf{+126}\%)$ | $803 \pm 65 \, (\mathbf{+110}\%)$ | $956 \pm 26 \, (\mathbf{+150}\%)$ |
| | hard | $20 \pm 10$ | $352 \pm 58 \, (\mathbf{+1660}\%)$ | $385 \pm 45 \, (\mathbf{+1825}\%)$ | $822 \pm 24 \, (\mathbf{+4010}\%)$ |
| Average | train | $543$ | $781 \, (\mathbf{+44}\%)$ | $940 \, (\mathbf{+73}\%)$ | $938 \, (\mathbf{+73}\%)$ |
| | easy | $321$ | $647 \, (\mathbf{+102}\%)$ | $789 \, (\mathbf{+146}\%)$ | $906 \, (\mathbf{+182}\%)$ |
| | hard | $126$ | $362 \, (\mathbf{+187}\%)$ | $375 \, (\mathbf{+198}\%)$ | $747 \, (\mathbf{+493}\%)$ |

Table 2: Ablation study. Percentages indicate variations compared to vanilla SAC.

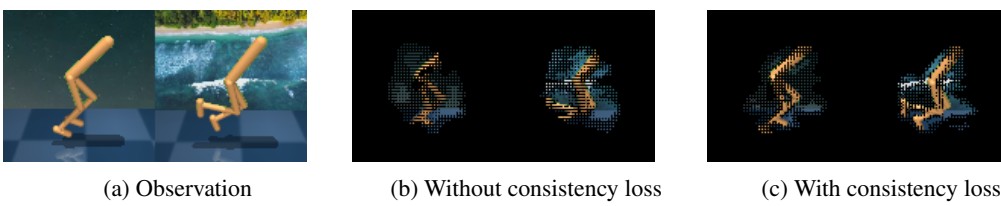

(a) Observation  (b) Without consistency loss  (c) With consistency loss

Figure 7: Comparison of SGQN attributions with and without consistency loss

that agents trained with our self-supervised learning objective obtain performance of the same order of magnitude as SODA agents (Table 1). Recall that SODA relies on the BYOL [Grill et al., 2020] self-supervised feature learning procedure, whose target labels differ notably from those proposed herein. The reported performance, compared to that of SODA suggests that attribution maps constitute a good labeling function that could be considered in the more general context of self-supervised learning. To a slightly lesser extent, adding the consistency loss to SAC also significantly improves its performance. The average score obtained improves by more than 44% on the training domain and by 102% and 187% respectively on the *video easy* and *video hard* domains. Similarly to the results obtained by Ismail et al. [2021] in supervised learning, the regularization of the critic's loss with the consistency term sharpens the attribution maps obtained (Figure 7). In SGQN, these fine-grained attributions provide higher quality labels to the auxiliary self-supervised learning phase, thus yielding significant performance improvements in training (+73% on average, range up to +242%), *video easy* generalization (+182% on average, range up to +399%), and *video hard* generalization (+493% on average, range up to +4010%).

## 5 Conclusion

The ability to filter out confounding variables is a long-standing goal in machine learning. For visual reinforcement learning, it is a pre-requisite for real-world deployment of learned policies, since we want to avoid at all costs situations where an agent makes the wrong decision due to distracting visual factors. In this work, we introduced saliency-guided Q-networks (SGQN), a generic method for visual reinforcement learning, that is compatible with any value function learning method. SGQN relies on the positive interaction of two core mechanisms of self-supervised learning and attribution consistency that jointly encourage the RL agent to be self-aware of the decisive factors that condition its value function. We implement an SGQN agent based on the soft actor-critic algorithm, and evaluate it on the DMControl generalization benchmark. This agent displays a dramatically more efficient learning curve than vanilla SAC on the various environments it is trained on. Most importantly, the policy it learns closes the generalization gap on environments that include confusing and distracting visual features, setting a new reference in terms of generalization performance. Since they rely on self-awareness of important pixels, SGQN agents are also very interpretable, in the sense that they provide a prediction of their own saliency maps, with no computational overhead.

The introduction of SGQN is an exciting milestone in RL generalization, and we wish to conclude this contribution by highlighting some limitations and perspectives for research that we believe are beneficial to the community. Attribution maps seem to be an efficient proxy for encouraging causal relationships within policies, but they are strongly grounded in an anthropomorphic point of view of what a visual policy should be. Extending their definition to more abstract notions of attribution is still a challenge and begs for important contributions, both theoretical and algorithmic. Similarly, such attribution maps appear to be relevant self-supervised learning targets, in order to learn good features for RL agents. Exploring whether this still holds for different tasks than RL is an open question. It is likely that one could design variations of SGQN that perform better than SG-SAC. The extension of SGQN agents to discrete action agents (e.g. DQN), or policy gradient methods (e.g. PPO) is a promising perspective in itself. Although generic and grounded in sound algorithmic mechanisms, the losses introduced by SGQN lack a formal connection to some measure of the generalization gap. Such a connection could provide insights to better self-aware, explainable agents with improved generalization capabilities. Finally, using SGQN as a building brick, among all those required to bridge the gap between simulation and real-world applications, is an exciting perspective.

## Acknowledgments

The authors acknowledge the support of the DEEL, ENVIA and MINDS projects, the funding of the AI Interdisciplinary Institute ANITI funding, through the French "Investing for the Future – PIA3" program under grant agreement ANR-19-PI3A-0004. This work benefited from computing resources from CALMIP under grant P21001.

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
