# OpenReview forum: "Look where you look! Saliency-guided Q-networks for generalization in visual Reinforcement Learning"
_NeurIPS.cc/2022/Conference — NeurIPS 2022 Accept_

### Official Review · Reviewer_S2Er · 2022-07-10

**Rating:** 5
**Confidence:** 5
**Soundness:** 3 good
**Presentation:** 4 excellent
**Contribution:** 3 good

**Summary:**

This paper proposes a saliency-guided Q-networks (SGQN) method that guides the policy to focus on important pixels for improving generalization sample efficiency. SGQN has two major components: 1) force the saliency of the augmented image to be close to that of the original image, 2) force the Q values to be close. The experiments show that it outperforms existing state-of-the-art methods on DMC-GB.

**Questions:**

1. The authors pick a large number for thresholding the binary mask. Why is this number selected? For a different task, do you need to choose a different number?
2. How does the proposed method compare with SODA+SVEA?
3. How does the proposed method perform on other benchmarks?

**Limitations:**

The authors mention their limitations as future research directions such as explanations and theoretical analysis of the proposed method. We appreciate the authors for being upfront.

**Strengths And Weaknesses:**

strengths:
1. This work shows that the learned policy could largely improve the generalization ability in the DMC-GB with self-supervised learning.
2. The learned policy could be interpretable with the use of saliency map.
3. The paper is well-written.

weaknesses:
1. The proposed method is only applied on DMC-GB. It is not sure that the proposed method is useful in other tasks other than the DMC-GB.
2. SGQN is a DrQ based algorithm since the batch size data is augmented by random shift. Then there are various places the authors can edit and use DrQ instead of SAC.
3. SODA+SVEA can be a stronger baseline since you implement two loss functions that are similar to the losses in the two papers respectively. To better illustrate the superiority, a comparison is preferred: the improvement does come from the saliency representation rather than any representation or embeddings with these two losses.
4. ‘One can note that agents trained with our self-supervised learning objective also outperform SODA agents (Table 1) on 4 out of 5 video easy environments and on 3 out of 5 video hard environments.’ This comparison result seems problematic according to the Table 1(SODA) and Table 2 (SAC+Self learning).
5. Missing references:

Fu, X., Yang, G., Agrawal, P., & Jaakkola, T. (2021, July). Learning task informed abstractions. In *International Conference on Machine Learning*
 (pp. 3480-3491). PMLR.

Yuan, Z., Ma, G., Mu, Y., Xia, B., Yuan, B., Wang, X., ... & Xu, H. (2022). Don't Touch What Matters: Task-Aware Lipschitz Data Augmentationfor Visual Reinforcement Learning. *arXiv preprint arXiv:2202.09982*.Chicago

---

> ### Author Response · Authors · 2022-07-29
> **Answer to reviewer S2Er**
>
> > The authors pick a large number for thresholding the binary mask. Why is this number selected? For a different task, do you need to choose a different number?
>
> This value was selected after a quick parameter search.
> It seems that (at least within the DMC), the 5% most salient pixels are sufficient to predict the value function and generalize.
> Although we currently do not discuss it in the paper, this is a (loose) measure of the amount of information actually present in the image and necessary for predicting the value.
> It is likely that one could exhibit worst case environments for which *all* pixels are necessary to predict the value.
> In such environments, SGQN might perform poorly.
> However, we argue that such environments are not representative of most realistic visual RL tasks, either real-world or simulated, where pixel information is very redundant (which is a cause for overfitting).
> Yet, it is likely that in other (maybe more difficult) environments (e.g. with more complex important objects moving across the screen), the necessary threshold for $\rho$ could be lower.
> We believe such a study and the possible automated adaptation of $\rho$ are a great perspective for future work.
> We propose to include this short discussion as an additional section in the appendix.
>
> In addition to this explanation, we are currently running experiments to report the efficiency of SGQN for a single value of $\rho$ across all environments.
>
> > The proposed method is only applied on DMC-GB. It is not sure that the proposed method is useful in other tasks other than the DMC-GB.
> > How does the proposed method perform on other benchmarks?
>
> We are doing our best to provide results on another benchmark.

---

> > ### Author Response · Authors · 2022-08-05
> > **Updated paper**
> >
> > We have updated the pdf file with all discussed modifications and new results. We draw your attention to the edits all along the paper, to the comparison with SODA+SVEA in Appendix F, to the experiment with a new environment in Appendix D, and to the extended discussion on the impact of initial saliency maps in Appendix G.

---

> > > ### Comment · Reviewer_S2Er · 2022-08-08
> > > **Response to rebuttal**
> > >
> > > The rebuttal addressed my concerns. I'll keep my rating.

---

> ### Author Response · Authors · 2022-07-29
> **Answer to reviewer S2Er**
>
> We thank the reviewer for this very thoughtful review. We provide an answer to all comments below.
>
> > SGQN is a DrQ based algorithm since the batch size data is augmented by random shift. Then there are various places the authors can edit and use DrQ instead of SAC.
>
> Thank you for this remark.
> We have based our implementation on that of Nicklas Hansen (https://github.com/nicklashansen/dmcontrol-generalization-benchmark) and it is true that the data augmentation of DrQ is still present in the training loop, even for SVEA.
> We are running the experiments without this data augmentation. So far, results seem similar to those obtained previously.
> Without this augmentation, our method no longer derives from DrQ.
> However, for the sake of comparison, we will add the results obtained by DrQ.
> As soon as the results are ready, we shall include them in the paper and report them here.
>
>
>
> > SODA+SVEA can be a stronger baseline since you implement two loss functions that are similar to the losses in the two papers respectively. To better illustrate the superiority, a comparison is preferred: the improvement does come from the saliency representation rather than any representation or embeddings with these two losses.
> > How does the proposed method compare with SODA+SVEA?
>
> Although we mostly agree with your point, we took a somehow different approach to this question.
>
> First, instead of combining SODA+SVEA, we performed the ablation study (line 307, table 2) to highlight which of the two mechanisms was more important.
> Even if each mechanism is useful in itself, it seems it is really the interplay between the two that yields the best improvements.
>
> Then, we would like to point out that, if the proximity with SODA (and BYOL) is clear, the comparison with SVEA is somehow more questionable since the "augmentation" of SGQN does not rely on an ad hoc procedure but rather on masking based on the gradient value (the saliency map).
> We agree the losses remain similar in spirit and have done our best to clearly attribute the deserved credit to SODA and SVEA, but we believe this difference is notable and hence it was somehow more informative to perform the ablation study than compare with SODA+SVEA.
>
> Also, we believe the combination of SODA and SVEA is actually yet another (very interesting) contribution.
> To the best of our knowledge this has not been studied in detail yet, so comparing against this combination seemed a bit out of scope for this paper.
>
> Nonetheless, we agree on the relevance of the combination and included it in our code.
> The experiments are currently running and we will report on their results as soon as possible (and include them in the paper).
>
> > ‘One can note that agents trained with our self-supervised learning objective also outperform SODA agents (Table 1) on 4 out of 5 video easy environments and on 3 out of 5 video hard environments.’ This comparison result seems problematic according to the Table 1(SODA) and Table 2 (SAC+Self learning).
>
> You are perfectly right!
> The mistake is fully on our side and we are sorry (and we actually don't understand how we reached this claim, we must have mixed values up with SGQN or between difficulty levels in the final rush).
> To make things completely explicit: SODA still outperforms SAC + saliency based self-supervised learning except on rare occasions (e.g. walker walk easy or walker stand hard).
> We believe predicting gradient information (such as saliency maps) remains a relevant target for learning good representations but, even though its score is of the same order of magnitude as SODA, it is clearly not as good as the process of SODA.
> We have rephrased the paragraph on the ablation study to lift any ambiguity.
> Fortunately, this does not change the main contribution and conclusions of the paper.
> We warmly thank you for pointing this mistake to us!
>
> > Missing references
>
> They are now included. Thank you for pointing them out to us!

---

### Official Review · Reviewer_1cpS · 2022-07-11

**Rating:** 5
**Confidence:** 3
**Soundness:** 3 good
**Presentation:** 2 fair
**Contribution:** 3 good

**Summary:**

The paper presents saliency-guided Q networks (SGQN), which use saliency maps of the Q network to guide learning. The goal is to focus attention on pixels that are relevant to the task, while ignoring irrelevant ones, to aid in visual generalization at test time. SGQN is evaluated on the DM Control generalization benchmark with video backgrounds, and compared with SAC, RAD, SODA, and SVEA.

**Questions:**

I don't think changing the quantile value ρ hyperparameter for different tasks is a fair comparison (Appendix table 3). This would require doing hyperparameter search for any new task encountered. For results, just report for the same value across all tasks.

L224: "relying on the assumption that the initial saliency maps are reasonably good." Why is this a good assumption? Some issues with saliency maps have been discussed in prior work, e.g. "Sanity Checks for Saliency Maps" from Adebayo et al.

Minor comments (no need to respond to these):
* I would not say that Deep RL has "outstanding efficiency in simulated visual control tasks" (L1).

**Limitations:**

There should be discussion of the limitations of the approach, which might talk about drawbacks of saliency maps, computational cost/speed, etc.

**Strengths And Weaknesses:**

Strengths
* Visual generalization in RL is an important problem.
* The "video hard" evaluation seems like a challenging test of generalization.

Weakness
* To me, the method seems cyclic. The saliency maps from the Q network are used to influence how the Q network is trained.
* Some of the baseline numbers don't seem to align with what was reported in prior work. For example, in Table 1, the finger spin (easy) performance for SODA is 535, but in the SODA paper it is 695 (higher than the 609 reported for SGQN). Similarly, SVEA is 537 (Table 1) vs 808 (original paper). These numbers are averaged across 5 random seeds, right?
* The paper is poorly organized. For example, Section 3 (the approach) has no subsections or paragraphs, making it hard to follow.

-------
Review update:
I thank the authors for their responses and additional experiments.

I looked in more detail at saliency-guided training (e.g. in Ismail et al.) and this technique seems to have some merit. My concern about the cyclic nature of the method was that it is only applying large gradients to the Q network and discarding small ones, which seems like it could result in unstable training and/or suboptimal use of the model parameters. Expanding the discussion of saliency-guided training (such as including the findings from Ismail et al. in related work) would help mitigate these concerns.

L293: “SGQN yields sharp saliency maps.” This evaluation in particular is cyclic since SGQN used this saliency technique in training while baselines did not. This point should be mentioned, or even better a different attribution technique should be used for this evaluation. See discussion of Goodhart’s Law in Ismail et al.’s OpenReview forum: https://openreview.net/forum?id=x4zs7eC-BsI

Paper organization: I would say the background and the method’s motivation should be separated out from related work. Regarding the method section (3), I find myself having to jump around quite a bit. For example, if I want to understand how the image augmentations are used, I must look at  L161, L177, L218, Figure 1, and Algorithm 1 (though τ is not actually used in the algorithm definition). Instead, the approach section could have clear subsections or paragraph labels for each learning phase (consistency and self-supervised objectives). The parts of the text and figure 1 could more clearly refer to each other. Algorithm 1 could probably be moved to the appendix since most of it does not relate specifically to the method.

My baseline numbers and ρ values questions have been resolved.

The primary reason for my low rating was the concern about the cyclic nature of the saliency-based training, which as I mentioned has mostly been resolved (since it has shown up in prior work). Thus I have increased my overall, soundness, and contribution ratings. I still encourage the authors to improve the discussion and organization of the text in the ways I have described.

---

> ### Author Response · Authors · 2022-07-29
> **Answer to reviewer 1cpS**
>
> > Some of the baseline numbers don't seem to align with what was reported in prior work. For example, in Table 1, the finger spin (easy) performance for SODA is 535, but in the SODA paper it is 695 (higher than the 609 reported for SGQN). Similarly, SVEA is 537 (Table 1) vs 808 (original paper). These numbers are averaged across 5 random seeds, right?
>
> We are surprised by this remark since the rest of the review seems to indicate the reviewer read the appendix (thank you for that effort!).
>
> Line 260 of the main text refers to Appendix A and B for complete hyperparameter details. Line 593 (Appendix B) in particular indicates "All scores were calculated on an average of 5 repetitions".
>
> Differences with the results of the original papers are discussed in Appendix C:
> "For the sake of simplicity and consistency, all experiments in Section 4 have been conducted with the same set of hyperparameters (except for the quantile value $\rho$, recalled in Table 4).
> A particular attention need to be paid to the number of times an action is repeated in the environments of the DMControl-GB, since it has an important influence on the scores reached by the agents.
> Hansen et al. [2021] indicate they run their experiments with an action repetition covering 4 time steps for all environments, except for Cartpole (8 time steps) and Finger spin (2 time steps).
> To avoid such heterogeneity across environments, we chose to report scores in Tables 1 and 2 with a constant action repetition parameter of 4 time steps for all environments.
> We repeated the experiments of Section 4 by setting the value of this parameter to 8 and 2 for the Cartpole and Finger spin environments respectively as suggested by Hansen et al. [2021].
> Table 5 reports the results obtained. [...]"
>
> We let the reviewer see for him/herself that we obtained results for SODA and SVEA in our experiments that are comparable to those of their respective original papers. In these configurations that are favorable to SODA and SVEA, SGQN outperforms them by an even larger margin (notably, +165% on finger spin hard).
> For clarity, we will add a reference to this appendix in the results section of the paper.
>
> > I don't think changing the quantile value ρ hyperparameter for different tasks is a fair comparison (Appendix table 3). This would require doing hyperparameter search for any new task encountered. For results, just report for the same value across all tasks.
>
> We agree that in the best of worlds, all algorithms should have a perfect set of preset hyperparameters that span all possible applications.
> In practice, we respectfully recall that many algorithms in the RL literature are often evaluated by searching for a good set of hyperparameters for each experiment.
> Nonetheless, we agree that it is a very desirable property to have reduced sensitivity to hyperparameter setting.
> To answer this remark, we first point out that the values of $\rho$ used in the experiments are very close to each other: 0.95 for walker walk, walker stand and finger spin, 0.98 for cartpole and ball in cup.
> We are currently running experiments to report SGQN scores for a constant $\rho=0.95$ and will include them in the paper.

---

> > ### Author Response · Authors · 2022-08-05
> > **Updated paper**
> >
> > We have updated the pdf file with all discussed modifications and new results. We draw your attention to the edits all along the paper, to the discussion on $\rho$ in Appendix E, and to the extended discussion on the impact of initial saliency maps in Appendix G.

---

> ### Author Response · Authors · 2022-07-29
> **Answer to reviewer 1cpS**
>
> We thank the reviewer for this feedback.
>
> We are however surprised by the discrepancy between the grading and the review. As far as we can tell, the reviewer did not find anything false in the paper and discards all the discussion and results obtained. Moreover, the formulated comments ("cyclic method", "missing subsections", "differences in empirical results"), if they can call for a discussion, do not seem to back the attributed grades (we understand a soundness of 1 as a grade for a paper that makes false claims, and a presentation of 2 as a paper that is purely unreadable). We believe there may be a misunderstanding and try to address all comments below.
>
> > To me, the method seems cyclic. The saliency maps from the Q network are used to influence how the Q network is trained.
>
> Indeed, the method is cyclic. We respectfully point out that it is the essence of bootstrapping: use one's own predictor to influence the learning targets. It is at the root of Q-learning (with strong theoretical justifications in the tabular case) and is a key idea of self-supervised learning methods such as BYOL (with more empirical evidence) which constitute the current state-of-the-art in self-supervised learning.
>
> We understand how this might be unsettling.
> Overall, and using the paper's notations, SGQN's critic update makes sure that the critic's output $Q_\theta(s,a)$ matches the classical target $r+\gamma Q(s,\pi_n(s))$ (this is the $L_Q$ part of the loss), and at the same time, the critic's output relies only on the pixels identified as informative, i.e. the output $Q_\theta(s,a)$ is the same as $Q_\theta(s\odot M_\rho,a)$ (this is the $L_C$ part).
> If the "oracle" indicating which pixels are informative is wrong, i.e. if too many confounding pixels are retained in $M_\rho$, then the first part of the loss still encourages that $Q_\theta$ is a solution to the Bellman equation.
> In that sense, $L_C$ acts as a regularizer (as indicated line 212): it allows discriminating between functions that would otherwise be equivalent approximate solutions to the Bellman equation.
> The relevance of the saliency maps (of the "oracle") in turn seems to stem from the quality of the representation $f_\theta$, similarly to the findings of the BYOL or SODA papers, themselves coming from the self-supervised learning procedure (of predicting one's own saliency map in the case of SGQN).
> Note that this prediction $M_\theta$ is *not* used in the critic update: only the features $f_\theta$ intervene when computing $M_\rho$.
>
> So, indeed, the method is "cyclic", but it is motivated and grounded, and we hope this clarifies things. We will update the paper to avoid possible confusions as best as we can.
>
> > L224: "relying on the assumption that the initial saliency maps are reasonably good." Why is this a good assumption? Some issues with saliency maps have been discussed in prior work, e.g. "Sanity Checks for Saliency Maps" from Adebayo et al.
>
> An alternate intuition on the mechanism of SGQN is to note first that initially, $\nabla_s Q_\theta$ is likely to assign random importance to all pixels, and thus $M_\rho$ focuses on random pixels spreadout uniformly throughout the image. So $s\odot M_\rho$ is comparable to a random uniform downsampling. Note also that many of the input image's pixels hold equivalent information to overfit $Q$, although not all will permit generalization. So this downsampling is unlikely to cause information loss. Then, few gradient steps on $L_Q$ are often enough to let the main "trends" of the value function emerge. At this stage, if the value of $Q_\theta$ itself is still inaccurate, if the features $f_\theta$ are descriptive enough, then $\nabla_s Q_\theta(s,a)$ start disambiguating between important and unimportant features and pixels which motivates the consistency loss term. What makes the $f_\theta$ features descriptive enough, is precisely the self-learning phase. So it seems that, overall, the interplay between the two phases of SGQN is mostly a virtuous circle, that relies on the assumption that the initial $M_\rho$ preserve enough information in $s$ to permit correct learning of $Q_\theta$. We will include this rephrasing in the paper to lift the ambiguity and explain why "the assumption that initial saliency maps are reasonably good" is reasonable.
>
> > The paper is poorly organized. For example, Section 3 (the approach) has no subsections or paragraphs, making it hard to follow.
>
> There are paragraph skips every ~10 lines, that correspond to transitions between ideas. We fail to see how adding subsections in a section that is 1.5 page long (in a 9 pages paper) and contains one figure and one algorithm would clarify anything. Since this is the only comment concerning the paper's presentation (which otherwise is rather classic), we express our bewilderment concerning the corresponding grade.

---

> ### Author Response · Authors · 2022-08-09
> **Additional answer to reviewer 1cpS**
>
> Thank you for these constructive comments.
> We now better understand your concerns and have updated the paper to address them and go along with your suggestions to clarify section 3.
>
> >I looked in more detail at saliency-guided training (e.g. in Ismail et al.) and this technique seems to have some merit. My concern about the cyclic nature of the method was that it is only applying large gradients to the Q network and discarding small ones, which seems like it could result in unstable training and/or suboptimal use of the model parameters. Expanding the discussion of saliency-guided training (such as including the findings from Ismail et al. in related work) would help mitigate these concerns.
>
> The consistency loss is applied in conjunction with the SAC critic loss computed on the non-augmented state.
> Hence, the full gradients are still used in the update of $Q_\theta$.
> The consistency objective acts as a regularizer allowing to discriminate between functions that would otherwise be equivalent approximate solutions to the Bellman equation.
> Moreover, the mask is only applied during the estimation of the Q-value, similar to the augmentations applied in SVEA.
> As reported in the SVEA paper, this does not lead to instability during the learning process, but rather has a stabilizing effect which can be visualized in our case on figure 3.
> We added this discussion within Appendix G and renamed it to "Discussion on the interplay between initial saliency maps and saliency guided training"
> We also added in a short sentence in the related work the findings from Ismail et al. as you suggested.
>
> > L293: “SGQN yields sharp saliency maps.” This evaluation in particular is cyclic since SGQN used this saliency technique in training while baselines did not. This point should be mentioned, or even better a different attribution technique should be used for this evaluation. See discussion of Goodhart’s Law in Ismail et al.’s OpenReview forum: https://openreview.net/forum?id=x4zs7eC-BsI
>
> Thank you for this remark, we totally agree.
> We added a comparison with other attributions methods in the appendix and added a reference to this appendix in the results section of the paper.
>
> > Paper organization: I would say the background and the method’s motivation should be separated out from related work. Regarding the method section (3), I find myself having to jump around quite a bit. For example, if I want to understand how the image augmentations are used, I must look at L161, L177, L218, Figure 1, and Algorithm 1 (though τ is not actually used in the algorithm definition). Instead, the approach section could have clear subsections or paragraph labels for each learning phase (consistency and self-supervised objectives). The parts of the text and figure 1 could more clearly refer to each other. Algorithm 1 could probably be moved to the appendix since most of it does not relate specifically to the method.
>
> We now better understand your impression of the organization of the paper.
> We have modified section 3 to clearly identify two paragraphs about the consistency and self-learning objectives.
> We have also edited Figure 1 to facilitate the identification of the two terms and added references to the figure in both paragraphs.
> The other changes that you suggested are unfortunately too cumbersome to be made before the end of the evening and the end of the discussion period.
> However, we commit ourselves to modify the paper in this direction in case it is accepted.
>
> We thank you again for these comments and hope these modifications will justify raising your overall evaluation.

---

### Official Review · Reviewer_zoMV · 2022-07-11

**Rating:** 7
**Confidence:** 3
**Soundness:** 3 good
**Presentation:** 3 good
**Contribution:** 3 good

**Summary:**

The paper introduces a generic saliency-guided method (SGQN) for visual reinforcement learning, aiming at improving the generalization capabilities and robustness of deep reinforcement learning policies. This is based on two mechanisms introduced by authors, both depending on masking the input image based on a computed attribution (saliency) map, and therefore inducing the agent to pay more attention where it is looking. The first consists in regularization of the value function with a consistency term (computed form Q values of original and masked input) that encourage the value function to depend mainly on pixels with high attribution. The second is exploiting data augmentation while introducing an auxiliary self-supervised learning objective where the agent trains to predict its own Q-value's saliency maps, based on features encoding information about pixels relevant to make decisions.

**Questions:**

- In Background and related work, when methods to improve generalization capabilities are presented, there are many acronyms that are never explicitly expressed. I think it would help readability writing the full name at least one.
- In the experimental comparison with other methods for generalization, and considering the background paragraphs, SAC and SODA belong to the Data augmentation approach. To which approach do RAD and SVEA belong (it is stated they all include data augmentation in one of their stages)? From my understanding, SGQN mix some aspect of data augmentation (which is used to generate the attribution mask), attribution and regularization methods. Could you say something about its advantages and limitations compared to methods that are not considered in the comparison, such as methods based on representation learning?
- How is the image augmentation function defined (at line 256 it says random overlay augmentation)? This means initial saliency maps are randomly generated? It seems to me that the mechanism works if initial saliency maps are properly defined, and therefore if there is a previous knowledge about which pixels are relevant to the task. What happens if the initial map is not good enough?
- In Fig. 6, I would make the caption more descriptive, specifying explicitly what is shown in each subplot.
- line 318: from Table 1 it seems SGQN outperforms SODA on 5 (not 3) out of 5 video hard environments.

**Limitations:**

Authors adequately address limitations and perspectives of their research in the Conclusion section. I share their view about the fact that the main limitation in the use of saliency maps to guide the RL learning process is that these strongly rely on a human point of view (and prior knowledge about the task).

**Strengths And Weaknesses:**

The paper overall quality is good and the contribution is significant. The context of the work is clearly illustrated with a proper literature review, and specific contributions are clearly described compared to existing approaches. While building up over existing methods, such as regularization and data augmentation, the paper introduces an original and significant architecture for saliency-guided RL. Differently from the majority of current approaches, in which saliency maps are used as tool for interpreting the RL agents behavior, this work incorporates them in the learning process.
The approach is experimentally validated and compared with state-of-the-art methods for generalization in continuous actions RL, demonstrating its significance in improving training efficiency, generalization capabilities, and providing a straightforward way for policy interpretability that does not require to compute the actual saliency map. Results are reported in a clear and complete way, and they are exhaustively discussed.
An advantage of the proposed method is that it can be combined with other techniques for attribution map computation and image augmentation, value function learning objective, and it is suited for both continuous and discrete action spaces, therefore fostering reusability.

---

> ### Author Response · Authors · 2022-07-29
> **Answer to reviewer zoMV**
>
> We thank the reviewer for these comments that will help us improve the paper.
>
> > In Background and related work, when methods to improve generalization capabilities are presented, there are many acronyms that are never explicitly expressed. I think it would help readability writing the full name at least one.
>
> We did our best to correct this.
>
> > In the experimental comparison with other methods for generalization, and considering the background paragraphs, SAC and SODA belong to the Data augmentation approach. To which approach do RAD and SVEA belong (it is stated they all include data augmentation in one of their stages)? From my understanding, SGQN mix some aspect of data augmentation (which is used to generate the attribution mask), attribution and regularization methods. Could you say something about its advantages and limitations compared to methods that are not considered in the comparison, such as methods based on representation learning?
>
> We have tried to correct the paper to avoid confusions.
> SAC does not perform any data augmentation.
> RAD does "raw" data augmentation on the input images.
> SVEA does careful data augmentation specifically on the critic of SAC.
> DrQ averages the predictions for a set of image transformations (image shifts) while learning the value function.
> SODA is very similary to BYOL, which is a representation learning method: it alternates between a representation learning phase (where features are trained to that random projections of the augmented input data can be reconstructed) and a value learning phase.
> In that sense, we have tried to highlight the proximity with SGQN.
> We believe your comment meets that of reviewer S2Er concerning the link to DrQ and we have tried to clarify the position of our contribution with respect to comparable approaches.
>
> > How is the image augmentation function defined (at line 256 it says random overlay augmentation)? This means initial saliency maps are randomly generated? It seems to me that the mechanism works if initial saliency maps are properly defined, and therefore if there is a previous knowledge about which pixels are relevant to the task. What happens if the initial map is not good enough?
>
> The augmentation function is genuinely that used by SODA [Hansen and Wang, 2021].
> We refer the reader to their paper for details but, in short, in consists in summing together the original image and one from the Places365 dataset, in random proportions.
> This does not cause initial saliency maps to be randomly generated.
> Saliency maps $M_{\rho}$ are computed from the original observation, thus they are not impacted by the augmented version.
> However, initial saliency maps are likely to look random since they represent the gradients of random functions close to zero (as per the classical initialization of neural networks).
> Consequently, when thresholded, these saliency maps are likely to yield pixels that are uniformly spread across the image.
> In turn, the masking operation originally acts as a random subsampling operation.
> Since many close pixels in the input image hold redundant information, the application of $M_\rho$ to input images is likely to preserve enough information to correctly predict the value function (minimize $L_Q$).
> As the $Q$-function becomes better and as the $f_\theta$ become better at predicting $M_\rho$, the saliency maps become sharper.
> We have updated the paper to better reflect this mechanism.
>
> > line 318: from Table 1 it seems SGQN outperforms SODA on 5 (not 3) out of 5 video hard environments.
>
> You are right!
> But actually our formulation was not that good on line 318.
> As reviewer S2Er pointed out, SAC with the self-supervised learning objective, although it is close in performance to SODA, does not outmatch SODA.
> SGQN does, of course, outperform SODA (by a large margin) but on line 318 we were discussing the ablation study.
> We have rephrased this paragraph to correct this mistake.
> Thank you for your vigilance.

---

> > ### Author Response · Authors · 2022-08-05
> > **Updated paper**
> >
> > We have updated the pdf file with all discussed modifications and new results. We draw your attention to the edits all along the paper and to the extended discussion on the impact of initial saliency maps in Appendix G.

---

> ### Comment · Reviewer_zoMV · 2022-08-09
> **Response**
>
> I thank the authors for their answers. I keep my rating.

---

### Author Response · Authors · 2022-08-02
**Paper update**

We thank all the reviewers for their valuable feedback. We have put in a major effort to address all their comments, questions and concerns, and we believe it has brought the paper to a much better level. This has strengthened the results and the claims made in the paper.

All major modifications in the pdf file have been highlighted in red in order to ease the reading.
Major changes (details in individual answers to the reviewers below) as per the reviewers requests:
- Added an appendix discussing the value of $\rho$ (the parameter thresholding the binary mask) and presenting the results when using the same value for all environments.
- Added DrQ in the benchmarking results.
- Added an appendix with a comparison with SODA+SVEA environments.
- Added an appendix discussing the cyclic nature of SGQN and in particular the assumption about the reasonably good initial saliency maps.
- Corrections in the paper regarding the performance of the self-supervised learning objective compared with SODA
- Adding missing references.
- Clarifying the acronyms of other methods
- Added a requested experiment in a new environment.

We also tried to better highlight the three core benefits where SGQN vastly outperforms previous state-of-the-art results: improved learning curves in the training domain (both learning speed and final score), much better generalization in the hardest domains, and interpretability by design.

---

### Meta-Review · Area_Chair_u5NW · 2022-08-27

**Recommendation:** Accept
**Confidence:** Certain

**Metareview:**

The paper proposes a method for ignoring task-irrelevant background information for RL algorithms trained directly from visual inputs. The method consists of two parts: (a) a consistency loss; (b) a self-supervised learning objective on masks. Reviewers unanimously agree that contributions made by the paper are significant, and I agree. The authors should address the remaining minor concerns of the reviewers about the presentation and keep the promises made in the rebuttal in the camera-ready version.

**Award:**

No

---

### Decision · Program_Chairs · 2022-09-14

Accept